# Redox-Cycling “Mitocans” as Effective New Developments in Anticancer Therapy

**DOI:** 10.3390/ijms24098435

**Published:** 2023-05-08

**Authors:** Rumiana Bakalova, Dessislava Lazarova, Akira Sumiyoshi, Sayaka Shibata, Zhivko Zhelev, Biliana Nikolova, Severina Semkova, Tatyana Vlaykova, Ichio Aoki, Tatsuya Higashi

**Affiliations:** 1Department of Molecular Imaging and Theranostics, National Institutes for Quantum Science and Technology (QST), Chiba 263-8555, Japan; 2Faculty of Medicine, Sofia University, St. Kliment Ohridski, 1407 Sofia, Bulgaria; 3Faculty of Medicine, Trakia University, 6000 Stara Zagora, Bulgaria; 4Institute of Biophysics and Biomedical Engineering, Bulgarian Academy of Sciences, 1113 Sofia, Bulgaria

**Keywords:** quinones, ascorbate, cancer, mitochondria, redox-cycling, prenylation, oxidative stress

## Abstract

Our study proposes a pharmacological strategy to target cancerous mitochondria via redox-cycling “mitocans” such as quinone/ascorbate (Q/A) redox-pairs, which makes cancer cells fragile and sensitive without adverse effects on normal cells and tissues. Eleven Q/A redox-pairs were tested on cultured cells and cancer-bearing mice. The following parameters were analyzed: cell proliferation/viability, mitochondrial superoxide, steady-state ATP, tissue redox-state, tumor-associated NADH oxidase (tNOX) expression, tumor growth, and survival. Q/A redox-pairs containing unprenylated quinones exhibited strong dose-dependent antiproliferative and cytotoxic effects on cancer cells, accompanied by overproduction of mitochondrial superoxide and accelerated ATP depletion. In normal cells, the same redox-pairs did not significantly affect the viability and energy homeostasis, but induced mild mitochondrial oxidative stress, which is well tolerated. Benzoquinone/ascorbate redox-pairs were more effective than naphthoquinone/ascorbate, with coenzyme Q0/ascorbate exhibiting the most pronounced anticancer effects in vitro and in vivo. Targeted anticancer effects of Q/A redox-pairs and their tolerance to normal cells and tissues are attributed to: (i) downregulation of quinone prenylation in cancer, leading to increased mitochondrial production of semiquinone and, consequently, superoxide; (ii) specific and accelerated redox-cycling of unprenylated quinones and ascorbate mainly in the impaired cancerous mitochondria due to their redox imbalance; and (iii) downregulation of tNOX.

## 1. Introduction

The concept of “epigenetic priming” of cells that is under mitochondrial control has revived interest in mitoepigenetic regulation of cancer [1]. Damaged mitochondrial checkpoints and redox imbalance have been shown to cause metabolic reprogramming in the nucleus via reversible or irreversible changes in the methylation and/or acetylation of nuclear genome [1]. In this regard, the development of pharmaceutical strategies to affect mitochondrial function and retrograde (mitochondrial-to-nuclear) signaling pathways offer great potential and hope for effective anticancer therapy.

“Mitocans” is a an acronym derived from the terms “mitochondria” and “cancer” [2]. This is a group of anticancer drugs whose anticancer activity is due to targeting mitochondria and disrupting their energetic and synthetic functions. The goal of this therapeutic approach is to reduce the cancer cells’ supply of energy and metabolites, that are crucial for their growth and proliferation, as well as to suppress and disrupt the mechanisms of mitochondrial-nuclear interplay. “Mitocans” could affect the viability of cancer cells through various mechanisms, targeting: (i) mitochondrial electron-transport chain (ETC) and oxidative phosphorylation (OXPHOS); (ii) tricarboxylic acid (TCA) cycle; (iii) redox-regulating enzymes; (iv) mitochondrial tumor-suppressor and apoptotic signaling pathways; (v) mitochondrial fission; (vi) mitochondrial DNA (mtDNA), and others [3]. This makes “mitocans” promising new developments in anticancer therapy. In general, their anticancer effect is accompanied by a significant alteration of the cellular redox-state, overproduction of reactive oxygen species (ROS), and induction of severe oxidative stress, which is detrimental to cancer. “Mitocans”, targeting the specific metabolism of impaired cancerous mitochondria, affecting redox imbalance, and inducing oxidative stress in cancer cells, but not in normal cells and tissues, may offer hope for selective and tolerable cancer treatment. In this context, redox-cycling “mitocans”, such as quinone/ascorbate (Q/A) redox-pairs, are attractive candidates.

It is generally accepted that the Q/A redox-pair induces oxidative stress and subsequent replicative stress in transformed cells [4,5,6,7,8] by targeting their impaired mitochondria and inducing local redox-cycling, accompanied by overproduction of ROS [9,10,11]. Quinones are also known to directly affect mitochondrial respiration and studies provide explanations for the molecular mechanisms of this mitochondrial interference [12,13,14]. More intriguing and attractive is that some of the Q/A combinations can discriminate cancer cells and tissues from normal ones, exerting a targeted anticancer effect.

How do redox-cycling “mitocans” recognize cancer cells from normal cells?

Recently, we proposed a hypothetical mechanism of the selective anticancer effect of Q/A redox-pairs, based on experiments with menadione/ascorbate (K3/A)-treated cancer cells and comparing them with normal cells of the same origin [9] (Figure 1). Menadione is also known as pro-vitamin K3. The hypothesis proceeds from the assumption that cancer cells are over-loaded with reducing equivalents, such as NADH and oncometabolite succinate, as well as over-charged due to the high Q10H_2_/Q10 ratio [15,16,17,18]. We found that K3/A decreases mitochondrial membrane potential and increases the level of mitochondrial superoxide. Overproduction of superoxide requires electrons coming from reducing equivalents. This leads to the depletion of succinate and NADH in mitochondria. At high doses of K3/A, these processes lead to mitochondrial collapse and cell death. At low/tolerable doses of K3/A, these processes could increase the sensitivity of cancer cells to conventional anticancer drugs, radiation therapy, and the immune system [9]. The mitochondria of normal cells are not over-loaded with reducing equivalents and over-charged, which hampers the redox-cycling of the two substances in them. Thus, the K3/A redox-pair could recognize cancer cells, exerting its targeted anticancer effect. We believe that a similar mechanism is valid for other Q/A combinations.

Some Q/A redox-pairs are also characterized by multiple beneficial effects: (i) pronounced synergistic cytotoxicity towards cancer cells, but not towards normal cells at the same doses [4,5,6,7,8,9,19]; (ii) suppression of tumor growth in vivo [4,9,20]; (iii) suppression of colony formation and tumor invasion [4,6,20]; (iv) anti-inflammatory and immunomodulating effects [6,9]; (v) potentiation of the effectiveness of chemotherapy and radiation therapy [2,9].

Naphthoquinone K3 and benzoquinone coenzyme Q are some of the most attractive quinones because of their vital functions and essentiality to the organism. So far, studies have mostly focused on the K3/A redox-pair. Published clinical trials of oral administration of K3/A have demonstrated its safety and effectiveness in humans [21,22]. This is the first promising step for its transfer to the clinic.

Coenzyme Q is a key component of the mitochondrial ETC, where it acts as an electron transporter. It also links the ETC to other metabolic pathways within the mitochondria, such as pyrimidine synthesis, fatty acid beta-oxidation, and amino acid catabolism, as well as to metabolic pathways outside the mitochondria [23]. Coenzyme Q is present in all cell membranes and lipoproteins and has multiple functions. Coenzyme Q10 (CoQ10) deficiency is linked to a variety of diseases such as myopathy, retinopathy, neuropathy, nephropathy, liver dysfunction, endocrine disorders, etc. [24,25]. All these abnormalities are accompanied by impaired mitochondrial respiration in the respective tissue, as well as by a crash of the antioxidant defense system. CoQ10 supplementation is fundamental to the treatment of patients with CoQ10 deficiency. CoQ10 and its analogues are also effective in treating patients with mitochondrial dysfunction not associated with CoQ10 deficiency [26]. This indicates that the ability to restore electron flow in the ETC and/or increase mitochondrial antioxidant capacity are important factors to their therapeutic potential. Overloading and overcharging of the CoQ “pools” in the mitochondria has recently been reported to be one of the main triggers of their dysfunction, cellular transformation, and carcinogenesis [18,27]. However, cancer patients rarely respond to CoQ10 supplementation, and the treatment has not been approved as effective [28].

K3 and the pharmacological ascorbate have also been found to interfere directly with the mitochondrial ETC, bypassing complex I and complex III deficiency [12,14,29,30]. The two substances are applied as a dietary supplement in combination with CoQ10 in the treatment of mitochondrial diseases [31].

It is interesting to note that the strong synergistic antiproliferative and cytotoxic effects on cancer cells are inherent for the combination of ascorbate with K3, but not for the combination of ascorbate with vitamins K1 or K2 [9]. K3 has also been found to be more effective than vitamins K1 and K2 when administered alone [9,32,33]. This suggests that prenylation of quinones abolishes their anticancer activity. This could explain, at least in part, the lack of effectiveness of CoQ10 in the treatment of cancer. Unprenylated quinones appear to have greater potential as anticancer agents than prenylated ones. However, this assumption needs further experimental verification.

Studies also suggest that ascorbate should not be considered simply as a pro-oxidant or antioxidant [18,27]. Ascorbate is one of the most abundant cytosolic redox-active compounds and could serve as a “buffer” of excess reducing equivalents in the cytoplasm of cancer cells, due to their oxidative environment. Steady-state levels of ascorbate are significantly higher in cancer cells compared to normal cells due to overexpression of vitamin C transporters (GLUT1, SVCT1, SVCT2) [34]. This is also a prerequisite for the targeted anticancer effect of the Q/A redox-pairs.

K3, coenzyme Q, and ascorbate are recognized as vital regulators of cellular redox-state and redox signaling that appear to determine cell fate—senescence or proliferation. Benzoquinones are believed to have better potential as anticancer agents than naphthoquinones [6,35]. However, not all quinones studied show synergistic cytotoxicity with ascorbate in cancer cells as found for K3. Many synthetic benzo- and naphthoquinones have also not been studied on normal cells and tissues, which does not allow one to assess their bioavailability and tolerability, especially in combination with ascorbate.

The aim of the present study is: (i) to elucidate the effectiveness of Q/A redox-pairs (containing CoQ and K3 analogues) as anticancer combination drugs, and potential tolerance to normal cells and tissues as a result of their redox-cycling in impaired mitochondria; (ii) to compare the effects of unprenylated versus prenylated quinones and to assess the impact of prenylation on selective Q/A-mediated anticancer effects; (iii) to compare the effects of benzoquinone/ascorbate versus naphthoquinone/ascorbate redox-pairs in vitro and in vivo, considering which of the two combinations is more efficient, tolerant, and promising for translational research in the future. Eleven Q/A redox-pairs were investigated, and the experiments were performed under identical conditions on cultured cells and cancer-bearing mice.

## 2. Results

### 2.1. Effects of Q/A Redox-Pairs on Cell Proliferation and Viability In Vitro

The names and chemical formulas of the investigated quinones are given in Figure 2A,B. Two of the selected benzoquinones were prenylated—CoQ1 (with one isoprene unit) and CoQ10 (with 10 isoprene units), and two of the naphthoquinones were prenylated—K1 and K2 (both with 4 isoprene units—saturated and unsaturated, respectively). Other benzo- and naphthoquinones were unprenylated. K3 is a natural metabolite in animals and humans—a catabolic product of vitamin K1 in the intestine and a circulating precursor of vitamin K2 [36]. Some of the selected quinones are bioactive compounds found in non-toxic plant products such as the black cumin (ThymoQ), medicinal mushrooms (CoQ0), and fermented wheat germ (DMBQ and MMBQ), suggesting their safety for humans [37,38,39,40]. They all are coenzyme Q analogues. BHQ is considered an improved K3 analogue [41]. AtovaQ is a naphthoquinone known to interfere with mitochondrial electron transport, most likely by inhibiting the operation of complexes II and III [10]. It is FDA approved as an anti-malarial drug.

The effects of Q/A redox-pairs on the proliferative activity of leukemic lymphocytes (Jurkat) are presented in Figure 2C. The following trends were observed:(1)The antiproliferative effects of benzoquinone/ascorbate combinations were more pronounced than that of naphthoquinone/ascorbate combinations (Figure 2C(a,b,c) versus Figure 2C(d,e,f)).(2)The Q/A redox-pairs containing unprenylated benzoquinones exhibited marked cytotoxicity within 48 h of incubation (Figure 2C(c)). In this case, the number of live cells in the Q/A-treated samples dropped below the initial level at: 3/300 μM/μM for CoQ0/A; 5/500 μM/μM for CoQ0/A and MMBQ/A; and 10/1000 μM/μM for all unprenylated benzoquinones. In the naphthoquinone group, K3/A and BNQ/A also exhibited cytotoxicity, but only at 10/1000 μM/μM (Figure 2C(d)).(3)The antiproliferative effect of the Q/A redox-pairs containing prenylated quinones (CoQ10, K1, K2) was negligible, while that of containing unprenylated quinones (CoQ0, K3, DMBQ, MMBQ, BNQ, ThymoQ) was markedly expressed even at low concentrations of Q/A such as 2/200 μM/μM and 3/300 μM/μM. In the coenzyme Q group, the shortening of the isoprenoid chain increased the antiproliferative activity in the order: CoQ0/A > CoQ1/A > CoQ10/A, which was very well seen at 10/1000 and 20/2000 μM/μM (Figure 2C(a,b),D).(4)Antiproliferative and cytotoxic effects were most pronounced for the CoQ0/A and DMBQ/A.

The next stage of the study aimed to clarify which of the selected quinones manifest a synergistic antiproliferative effect with ascorbate (Figure 2D). The experiments were performed on leukemic lymphocytes treated with ascorbate and quinone alone and in combination. ThymoQ/A, DMBQ/A, MMBQ/A, CoQ0/A, K3/A, and BNQ manifested synergistic antiproliferative effects (Figure 2D(b,c,d,e,h); Appendix A). In this case, the synergism between quinone and ascorbate was well-expressed at concentrations between 1/100 and 5/500 μM/μM, but not at higher concentrations (Figure 2D(l); Appendix A). The effects of combinations of ascorbate with AtovaQ or prenylated quinones did not differ from that of the corresponding quinone administered alone (Figure 2D(f,g,i,j,k); Appendix A).

The IC50 values confirm the above conclusions (Figure 2E). The antiproliferative activity and cytotoxicity of the selected Q/A redox-pairs decreased in the order: CoQ0/A > DMBQ/A > ThymoQ/A > MMBQ/A > BNQ/A > K3/A > AtovaQ/A > prenylated quinones (CoQ1, CoQ10, K1, K2). IC50 value for CoQ0/A was 2.2/220 μM/μM versus >20/2000 μM/μM for its prenylated analogues CoQ1 and CoQ10. IC50 value for M/A was 7.5/750 μM/μM versus >20/2000 μM/μM for its prenylated analogues K1 and K2.

CoQ0/A, DMBQ/A, MMBQ/A, K3/A, and ThymoQ/A did not affect the viability of normal lymphocytes at concentrations up to 20/2000 μM/μM (Figure 2F). AtovaQ/A and BNQ/A demonstrated relatively weak but significant cytotoxicity to normal lymphocytes at a concentration of 20/2000 μM/μM (*p* < 0.05) (Figure 2F).

Based on the data described above, CoQ0/A was distinguished as the redox-pair with the most pronounced antiproliferative and cytotoxic effects, especially in leukemic lymphocytes. Its antiproliferative activity and cytotoxicity was investigated on seven additional cell lines, comparing the effects on cancer and normal cells of the same origin (Figure 2G). CoQ0/A strongly suppressed the proliferation and decreased the viability of colon cancer cells (Colon26), breast cancer cells (MCF7), and glioblastoma cells (U87MG, GS9L) without significantly affecting the viability of normal colon epithelial (FHC), normal breast epithelial (MCF10A) and normal microglial cells (EOC2) in the doses tested so far.

### 2.2. Effects of Q/A Redox-Pairs on Mitochondrial Oxidative Stress and Steady-State ATP In Vitro

The next stage of our study aimed to elucidate the effect of Q/A redox-pairs on mitochondrial redox-state and cellular energy balance, analyzing two parameters: mitochondrial superoxide and steady-state ATP.

Combinations of ascorbate with ThymoQ, DMBQ, MMBQ, CoQ0, and K3 (all unprenylated quinones) induced a dose-dependent overproduction of mitochondrial superoxide in cancer cells: ~four–six-fold above the baseline recorded in untreated cells (Figure 3A). CoQ1/A and BNQ/A also induced mitochondrial superoxide production, but the effect was less pronounced. All long-chain prenylated quinones did not affect the level of mitochondrial superoxide, and AtovaQ/A even decreased it ~two-fold compared to the control. The highest level of mitochondrial superoxide was recorded in the CoQ0/A-treated cells, followed by DMBQ/A-treated (Figure 3A). In normal lymphocytes, both redox-pairs increased mitochondrial superoxide ~1.5–2-fold in a concentration-independent manner (Figure 3B) and this effect appeared to be well tolerated, given the lack of cytotoxicity, as in K3/A-treated normal cells shown in our previous study [9].

ThymoQ/A, DMBQ/A, and CoQ0/A induced a very strong depletion of ATP in leukemic lymphocytes even at concentration of 3/300 µM/µM (Figure 3C). The steady-state ATP level has dropped over 90% compared to the baseline recorded in untreated cells. MMBQ/A, CoQ1/A, K3/A and BNQ/A also significantly decreased ATP in cancer cells, although their effect was less pronounced (Figure 3C). The combinations of ascorbate with prenylated quinones (CoQ10, K1, K2) and AtovaQ in concentrations of 3/300 and 5/500 µM/µM did not affect the level of ATP, while in a concentration of 10/1000 µM/µM they decreased ATP by 60% in leukemic lymphocytes. In normal lymphocytes, DMBQ/A and CoQ0/A did not significantly affect the level of ATP, and the same was valid for K3/A (Figure 3D).

Very good correlations were established between the analyzed parameters: cell proliferation/viability and mitochondrial superoxide (R = −0.8537; *p* < 0.001) (Figure 3E); cell proliferation/viability and steady-state ATP (R = +0.8197; *p* < 0.05) (Figure 3F); and mitochondrial superoxide and steady-state ATP (R = −0.6982; *p* < 0.001) (Figure 3G). The data described above show that overproduction of mitochondrial superoxide is accompanied by ATP depletion, suppression of proliferation, and decrease of cell viability.

Three Q/A redox-pairs were selected based on their effect on mitochondrial superoxide: (i) CoQ0/A—inducing the highest mitochondrial superoxide production among the selected benzoquinone/ascorbate combinations; (ii) K3/A—inducing the highest mitochondrial superoxide production among the selected naphthoquinone/ascorbate combinations; and (iii) AtovaQ/A, which did not increase the level of mitochondrial superoxide above that in untreated cells. The effects of CoQ0, K3, AtovaQ, and ascorbate administered alone and in combination were also analyzed in other cancer cell lines (Figure 3H,I,J). This experiment aimed to elucidate whether the effects of these Q/A redox-pairs on mitochondrial superoxide are universal or not, as well as to clarify the mechanism of mitochondrial interference. CoQ0/A and K3/A were found to induce overproduction of mitochondrial superoxide in Colon26, MCF7, and U87MG cells (Figure 3H), while AtovaQ/A significantly decreased the level of mitochondrial superoxide in Colon26 and MCF7 but did not affect this parameter in U87MG cells. The effects of K3 and ascorbate administered alone were negligible, whereas CoQ0 alone significantly increased mitochondrial superoxide in all cancer cell lines.

### 2.3. Effects of Q/A Redox-Pair on Tumor Growth and Tissue Redox-State In Vivo

In vitro experiments clearly indicated that CoQ0/A possessed the most pronounced anticancer effects among the selected Q/A redox-pairs. CoQ0/A was further investigated for suppression of tumor growth on cancer-bearing mice and the effect was compared to that of K3/A.

The experiments were performed on mice with glioblastoma and colon cancer hind paw xenografts. Tumor-bearing mice were treated with the respective drug using serial topical subdermal (s.d.) injections.

In the first cancer model, glioblastoma cells were transplanted into the hind paw and when the tumor size was approximately 25 mm^3^, the mice were divided in 2 groups: (i) single s.d. injection of CoQ0/A (1× CoQ0); (ii) 2 s.d. injections of CoQ0/A with a 3-day- interval between injections (2× CoQ0); and (iii) control—single s.d. injection of saline solution (Appendix A). Tumor growth was analyzed macroscopically using a caliper, as well as T_2_W MRI for more accurate measurements on selected animals.

CoQ0/A significantly decreased tumor growth compared to the control group within 25 days after cell transplantation and 18 days after drug administrating, regardless of the number of injections (Figure 4A). However, on day 32, the tumor entered a logarithmic phase of growth in mice receiving a single dose of the drug (Figure 4A). In this case, there was no significant difference in tumor size between the control group and the 1× CoQ0/A treated group on day 45 (Figure 4A). In mice injected twice with the drug within one week, the tumor grew slowly until day 45 after cell transplantation and the tumor size was ~7-fold lower that in control mice (~32 ± 6 mm^3^ versus 219 ± 48 mm^3^, respectively) (Figure 4B). This is also illustrated on the 3D magnetic resonance images (Figure 4C).

Similar data were obtained on the colon cancer xenograft model and s.d. injection of CoQ0/A or K3/A near the tumor (Figure 4D). Colon cancer cells were transplanted into the hind paw and when the tumor size was approximately 120 mm^3^, the mice were divided into 3 groups: (i) CoQ0/A-treated; (ii) K3/A-treated; and (iii) control (Appendix A). Drug-treated mice received six s.d. injections (twice per week), while control mice were injected with saline. Tumor growth was analyzed macroscopically using a caliper. CoQ0/A and K3/A significantly decreased tumor growth compared to the control group. Suppression of tumor growth was more pronounced in CoQ0/A-treated mice than in K3/A-treated mice. Moreover, 6× Q0/A decreased tumor size from ~120 mm^3^ before treatment to ~45 mm^3^ on day 35 after transplantation and day 26 after the first injection of the drug (Figure 4D). In both drug-treated groups, survival was significantly higher compared to the control group and the median survival of CoQ0/A-treated mice was longer compared to K3/A-treated mice (Figure 4E,F). Both drugs significantly decreased the tissue-reducing capacity and tNOX expression in the tumor as analyzed ex vivo using TAC/TRC assay or ELISA, respectively (Figure 4F). The effect of CoQ0/A on TAC level was more pronounced than that of K3/A, indicating a higher level of oxidative stress in the tumor of CoQ0/A-treated mice.

### 2.4. Summary of the Main Experimental Findings

The experimental results outline three well-defined trends:(1)Q/A redox-pairs containing unprenylated quinones were characterized by pronounced antiproliferative and cytotoxic effects on cancer cells. This was accompanied by a dose-dependent overproduction of mitochondrial superoxide and accelerated ATP depletion. Q/A redox-pairs containing prenylated quinones, especially long-chain ones, did not significantly affect cancer cell proliferation and viability, as well as mitochondrial oxidative stress and energy balance. Their effects on these parameters were negligible.(2)Q/A redox-pairs containing quinones such as CoQ0, DMBQ, MMBQ, and K3 did not significantly affect the viability of normal cells and steady-state level of ATP in them. These redox-pairs induced mild and dose-independent mitochondrial oxidative stress in normal cells, which seems to be well tolerated.(3)The anticancer effects of the investigated benzoquinone/ascorbate redox-pairs were more pronounced than those of naphthoquinone/ascorbate redox-pairs. The most effective was CoQ0/A followed by DMBQ/A, as both redox-pairs are a better alternative to the widely investigated K3/A.(4)CoQ0/A was more effective than K3/A in suppressing tumor growth and increasing survival in the investigated mouse models (glioblastoma and colon cancer xenografts). CoQ0/A decreased the size of the tumor, while K3/A caused only a growth arrest (*p* < 0.05). Both redox-pairs did not induce adverse drug-related side-effects such as fever, tetraplegia, convulsions, etc., that are characteristic of many conventional anticancer drugs.

## 3. Discussion

One of the most significant and potentially applicable findings in our study was that prenylation of quinones is essential for their tolerance to normal cells. However, prenylation abolished their targeted anti-proliferative and cytotoxic effects to cancer cells, when quinone was administered alone or in combination with ascorbate. We hypothesized that expression and activity of prenyltransferases is one of the main factors regulating and ensuring the selectivity of Q/A redox-pairs to cancer, especially those containing CoQ0, K3, and their analogues with high structural similarity.

In 2010, Nakagawa et al. have reported that prenylation of K3 to menaquinone (K2) is mediated by the mitochondrial UbiA prenyltransferase domain containing protein 1 (UBIAD1), also known as transitional epithelial response protein 1 (TERE1) [42]. In 2013, the same enzyme was found to be involved in the synthesis of CoQ10 in the Golgi membrane compartment [43]. No other enzymes related to CoQ10 synthesis were found in these organelles. This allows us to assume that UBIAD1 may be involved in the prenylation of CoQ0 or another similar precursor of CoQ10. This prenyltransferase is downregulated in multiple cancers [44], which is a prerequisite for the targeted anticancer effect of unprenylated quinones, especially with ascorbate, and their safety for normal cells and tissues. It is likely that this prenyltransferase is involved in the prenylation of DMBQ, MMBQ, ThymoQ, and/or BNQ in normal cells due to their structural similarity to CoQ0 or K3, respectively. The combination of these quinones with ascorbate also exhibited selective synergistic cytotoxicity to cancer cells and tolerance to normal cells of the same origin (Figure 2C,F). Recent cohort studies highlight UBIAD1 expression and activity as one of the newest prognostic markers in cancer [45,46]. UBIAD1 is known as a tumor suppressor [42,45,46]. Down-regulation of the enzyme has been shown to activate Ras-MAPK signaling [47] and mevalonate pathway, elevating cholesterol level [48]—decisive factors in carcinogenesis and cell survival [49].

Prenylated quinones are expected to reside in the lipid bilayer of biomembranes, including mitochondrial ones. Thus, prenylation should limit the mobility of the quinone in biomembranes and active sites of ETC complexes. This may explain, at least in part, the overproduction of mitochondrial superoxide caused by the unprenylated quinones and the lack of effect of the prenylated ones. The arguments for this statement are described below.

CoQ10 is an integral part of the mitochondrial ETC and CoQ analogs are known to interfere directly in the operation of complexes I and III [13,50,51]. Molecular docking and crystallographic analysis demonstrated that some naphthoquinones such as K3 and AtovaQ also have complementarity with the various CoQ binding sites on these mitochondrial complexes [10]. Thus, K3 can affect mitochondrial electron transport, but this does not apply to K2 and K1 in mammals [52]. Unprenylated/short-chain quinones and ascorbate are found to bypass complex I deficiency by injecting electrons directly to complex III and/or cytochrome c for ATP synthesis [53]. A key role in the destructive generation of superoxide has been attributed to the semiquinone form of CoQ10 in the Q_o_-pocket of complex III [54,55]. The mobile unprenylated quinone can replace the poorly mobile and over-reduced CoQ10 in the respiratory chain, and to be converted to a semiquinone. The unprenylated semiquinone can easily leave the active site of complex III and interact rapidly with molecular oxygen in the mitochondrial matrix, causing overproduction of superoxide. However, such a mechanism is unlikely to exist for prenylated quinones such as CoQ10, K1, and K2. For example, an ESR study shows that only K3 produces superoxide at physiological pH (7.4), whereas K2 and K1 are much less active or inactive as superoxide generators under the same experimental conditions [56]. Long-chain quinones are localized in the lipid bilayer and should have limited mobility in the mitochondrial membrane, as well as highly impeded redox-cycling with ascorbate localized in the water phase. The long isoprenoid chain does not allow the semiquinone to leave the active sites of the ETC complexes and easily and quickly interact with oxygen to produce a large amount of superoxide. We assume that cancerous mitochondria have “unbalanced”, over-reduced, and over-charged CoQ “pools” (Figure 1), and the above mechanism is intrinsic to them. The described mechanism is unlikely for normal cells with “balanced” CoQ “pools” and regular electron transport in the ETC. Moreover, prenylation of quinones to their long-chain analogs in normal cells will restrict the accelerated redox-cycling in mitochondria. In turn, this should significantly limit overproduction of ROS in Q/A-treated normal cells.

The most widely discussed mechanisms for overproduction of superoxide and hydrogen peroxide in Q/A-treated cells are: (i) non-enzymatic ascorbate-driven one-electron redox-cycling of quinone (Figure 5A), and (ii) enzyme-facilitated one-electron redox-cycling of unprenylated and short-chain quinones, catalyzed by cytochrome P450 oxidoreductase (CYP450), thioredoxin reductase, etc. (Figure 5B) [9,57,58]. Both mechanisms lead to production of semiquinone, which is subsequently oxidized non-enzymatically with the production of superoxide. These mechanisms are proposed to explain the synergism between K3 and ascorbate in cancer cells, but they do not explain the selectivity. There are serious arguments against the possibility that these two mechanisms dominate in cancer cells, which were detailed in our previous articles [9,27]. Briefly, in cells, unprenylated and short-chain quinones exist mostly in their reduced (enol) forms due to the presence of NAD(P)H:quinone oxidoreductase 1 (NQO1), catalyzing their two-electron reduction. This enzyme is up-regulated in cancer [58,59]. In cells, ascorbic acid also exists mostly in reduced form due to the activity of two enzymes: (i) Cyb5R3, which converts semi-dehydroascorbate to ascorbate by one-electron reduction, and (ii) glutathione peroxidase, which converts dehydroascorbate to ascorbate by two-electron reduction [60,61]. These enzymes are also found to be up-regulated in cancer [27,59,60,61]. We assume that the predominant existence of quinone and ascorbate in their reduced forms (quinol/ascorbate) excludes the interaction between both substances in real time, as well as the overproduction of ROS as a result of non-enzymatic or enzyme-facilitated one-electron redox-cycling (Figure 5A,B). In cells, one-electron redox-cycling mechanisms of quinone are possible only if the quinol is oxidized by additional reactions. We believe that one of these reactions is the oxidation of quinols to semiquinones in mitochondrial ETC, due to their similarity to CoQ10 (Figure 5C).

Our hypothesis is that the synergism between quinone and ascorbate, overproduction of mitochondrial ROS, and targeted anticancer effect are most likely due to: (i) suppression of quinone prenylation in cancer cells but not in normal cells, and (ii) conversion of quinol to semiquinone in the respiratory chain and highly specific and accelerated redox-cycling between the two molecules in impaired cancerous mitochondria (Figure 1 and Figure 5C).

Since CoQ0 has greater structural similarity to CoQ10 and a higher affinity for the mitochondrial ETC than K3, it should be more potent as inductor of mitochondrial superoxide, as found in our study (Figure 3A,H–J). The same is true for unprenylated CoQ0 analogues compared to unprenylated K3 analogues.

The data in Figure 3H–J show that the synergism between CoQ0 and ascorbate in mitochondrial superoxide production was less pronounced than between K3 and ascorbate. Based on these data, we suppose that the redox-cycling of CoQ0 and its unprenylated analogues is most likely directly coupled to the ETC and is not as strongly dependent on ascorbate, whereas the redox-cycling of K3 is more dependent on ascorbate. In both cases, we observed an induction of severe oxidative stress in cancerous mitochondria and ATP depletion (Figure 3).

In the case of AtovaQ, which is known to mimic CoQ structure and interfere with complexes II and III [10], we found an opposite effect on mitochondrial superoxide: (i) in some cell lines (Jurkat, Colon26, MCF7) this parameter decreased below the baseline of untreated cells, and (ii) in others (U87MG) it was not affected (Figure 3A,H–J). Molecular docking and crystallographic analysis have shown that AtovaQ bounds to Q_o_-site of complex III and its docked poses overlap with those of 1,4-naphthoquinone (K3 analogue) and CoQ1 in the same site [10]. Another favorable AtovaQ docking pose overlaps with docking pose of aptenin—inhibitor of complex II [10]. We established that the antiproliferative effect of AtovaQ was comparable to that of K3 and CoQ1, and much less pronounced than that of CoQ0 and its unprenylated analogues: ThymoQ, DMBQ, and MMBQ (Figure 2D(f,h,k)). However, no synergism was observed between AtovaQ and ascorbate (Figure 2D(k)). The anticancer effect of AtovaQ is most likely due to interference with complexes II and III, leading to inhibition of the TCA cycle and OXPHOS and, as a result, suppression of ATP production in cancer cells. However, our data indicated that the anticancer effects of AtovaQ- and AtovaQ/A were not associated with mitochondrial oxidative stress (Figure 3A,H–J). Other authors reported induction of oxidative stress in AtovaQ-treated cancer cells but at high concentrations (≥15 μM) [10,62]. Recently, Kapur et al. found that exposure of adenocarcinoma cells (ECC-1, OVCAR-3) to 25 μM of AtovaQ resulted in a ~80–90% increase of intracellular superoxide, analyzed by MitoSOX fluorescence [10]. Alharbi et al. analyzed the level of hydroperoxides in AtovaQ-treated cancer cells (15–30 μM), which was about two times higher than the baseline level recorded in untreated cells [62]. Regardless of methodological differences, as well as the different cancer cell lines used in the published articles, the reported effects of AtovaQ on intracellular ROS production are mild, compared to those of the unprenylated benzo- and naphthoquinones at the 10 μM concentration analyzed in our study (Figure 3A,H–J). One of the possible reasons that AtovaQ does not induce a significant oxidative stress in mitochondria could be the shielding of functional groups from the rings in its molecule, which would restrict the conversion of the quinol to semiquinone.

Another interesting finding in our study is that Q/A redox-pairs, especially those containing unprenylated quinones, induced severe ATP depletion in cancer cells but not in normal cells of the same origin (Figure 3C,D). This effect was most pronounced for CoQ0/A, DMBQ/A, and ThymoQ/A, followed by BNQ/A, MMBQ/A, CoQ1, and K3/A (Figure 3C). It should be noted that redox-pairs, containing long-chain quinones at concentration of 10 μM, also significantly decreased the level of ATP in the cancer cells.

ATP depletion in Q/A-treated cells is most likely due to inhibition of both OXPHOS and glycolysis. Cancer cells are characterized by metabolic flexibility and high glycolytic capacity, and can switch energy supply from one pathway to the other [63]. One of the most discussed theories, explaining the Q/A-induced ATP depletion in cancer cells, particularly K3/A-treated and CoQ0-treated, attributes this effect to the activation of poly-[ADP ribose] polymerase 1 (PARP1) and inhibition of glycolysis, as a result of an acute depletion of NAD^+^ [4,7,20,64,65]. This mechanism is also assigned to the oxidative stress induced by Q/A.

Our data demonstrate that benzoquinone/ascorbate redox-pairs are better inducers of oxidative stress and more effective anticancer agents, in vitro and in vivo, than naphthoquinone/ascorbate redox-pairs. Benzoquinones have a greater structural similarity to CoQ10 and a stronger affinity to the respiratory chain. Therefore, the more pronounced anticancer effect of benzoquinone/ascorbate redox-pairs compared to naphthoquinone/ascorbate redox-pairs supports our hypothesis of the mitochondrial nature of their anticancer activity. Below, we would like to highlight several recent articles supporting this assumption as well.

Despotovic et al. reported that mild oxidative stress, induced by tolerable concentrations of K3 (5–20 mM), triggers non-generalized and non-toxic autophagy in cells, which protects them [66]. However, ascorbate (0.5–2 mM) converts K3-induced autophagy from nontoxic to cytotoxic, which is accompanied by severe oxidative stress [66]. These experiments were performed in a human glioblastoma (U251) model using autophagy markers such as autophagosome-associated LC3-II and belcin-1 expression, and p62 degradation. Severe oxidative stress and activation of cytotoxic autophagy have also been recently reported in human glioblastoma (U87MG, GBM8401) and ovarian carcinoma (SKOV-3) cells treated with high concentrations of CoQ0 [38,65]. The authors analyzed the same autophagy markers. The combination of CoQ0 with ascorbate was not investigated in their study. It should be noted that CoQ0 administered alone induced cytotoxic autophagy at 5 mM in ovarian carcinoma, and 20 mM in human glioblastoma cells. Notably, the concentrations of CoQ0 inducing cytotoxic autophagy were lower than those of K3 mentioned above [38,65,66]. Therefore, doses that are harmless to cancer cells and tissues, in the case of K3, may be harmful in the case of CoQ0. This may explain, at least in part, the more pronounced effect of CoQ0/A on tumor growth in vivo compared to K3/A (Figure 4D). Repeated administration of CoQ0/A resulted in significant suppression of tumor growth in colon cancer-bearing mice with partial tumor resorption, whereas treatment with K3/A resulted in only tumor growth arrest (Figure 4D). In both cases, tumor growth resumed after ending of treatment—much slower in CoQ0/A-treated mice compared to K3/A-treated mice. A longer treatment with CoQ0/A may solve this problem. This is one of the challenges of any anticancer therapy and should be a subject of future research. Ex vivo analysis of the reducing capacity of tumor tissues, isolated from colon cancer grafted mice, indicated that CoQ0/A and K3/A induced oxidative stress in the tumor, and the level in CoQ0/A-treated mice was about twice as high as that in K3/A-treated (Figure 4F).

CoQ0, CoQ1, and K3 were found to induce selective/distinct mitochondrial-mediated cytotoxicity in cancer cells via inhibition of mtDNA polymerase-γ, while CoQ10, K1, and K2 did not affect mtDNA polymerase-γ activity and have a negligible effect on cancer cell viability [67,68]. CoQ0 was also found to induce mitochondrial permeability transition pore (PTP) opening, which triggers apoptosis via ROS-mediated VDAC1 upregulation in cancer cells (HL-2), as well as tumor growth suppression in cancer-bearing mice [69].

The downregulation of tNOX in the tumors of CoQ0/A- and K3/A-treated mice is also an interesting fact (Figure 4F), explaining the suppression of tumor growth and the increase of their survival (Figure 4D,E). tNOX has been found to be up-regulated in cancer cells, down-regulated in slow-proliferating non-cancer cells, and currently undiscovered in non-proliferating normal cells [70,71]. Thus, suppression of tNOX with anticancer agents such as CoQ0/A and K3/A could selectively inhibit cell growth and induce apoptosis in cancer cells, but not in normal cells. Some conventional anticancer drugs have been shown to transiently upregulate tNOX expression, thereby enhancing the migration of cancer cells and causing the development of drug resistance [72,73]. Up-regulation of tNOX has been found to correlate with a poor prognosis and low survival in patients with glioblastoma and colon cancer [70,71,74]. Inhibition of tNOX expression and/or activity was recently reported to affect mitochondrial function, increasing ROS-dependent mitochondrial autophagy and inducing apoptosis in cancer cells [75]. In this context, tNOX could be a valuable therapeutic target, distinguishing cancer cells from normal cells and enabling selective damage of cancerous mitochondria. tNOX is a hydroquinone (NADH) oxidase [76] and could also maintain the level of the oxidized forms of CoQ0 and K3 required for all redox-cycling mechanisms, including the redox-cycling with ETC and/or ascorbate in cancerous mitochondria (Figure 5).

It has been found that Q/A redox-pairs may influence the integrity of the tumor microenvironment and the activity of tumor-associated immune cells and fibroblasts. We recently reported that K3/A treatment significantly decreased tumor cell density and increased tumor perfusion, which is indirect evidence of its influence on the tumor microenvironment [11]. Similar effects were observed in CoQ0/A-treated glioblastoma mice—intracranial model (data will be published elsewhere). These findings were consistent with published histological data obtained on K3/A-treated cancer cells and tissues [77]. K3/A (at certain doses) has been found to cause a specific form of cell death called autoschizis, characterized by a reduction in cell size due to loss of cytoplasm by self-excision without loss of cell organelles, morphological degradation of the nucleus, and formation of apoptotic bodies [77]. CoQ0 and K3 have also been found to affect macrophage activity and inhibit inflammation by targeting the NLRP3 inflammasome [78,79]. K3/A suppressed PD-L1 expression in cancer cells [9].

In conclusion, our study proposes a pharmacological strategy for cancer treatment based on exploiting the unique difference between cancer cells and normal cells in their response to redox-cycling “mitocans” such as quinone/ascorbate (Q/A) redox-pairs. This strategy aims to distinguish and destroy impaired cancerous mitochondria by Q/A, which makes cancer cells fragile and vulnerable to the immune system and conventional therapies. Thus, it ensures highly selective mitochondria-mediated anticancer effects and tolerance to normal cells and tissues—one of the main goals of advanced anticancer therapy.

Q/A redox-pairs containing unprenylated quinones are attractive candidates as combination drugs in adjuvant anticancer therapy, but not Q/A redox-pairs containing prenylated long-chain quinones. Targeted anticancer effects of Q/A redox-pairs and their tolerance to normal cells and tissues are attributed to: (i) downregulation of quinone prenylation in cancer, leading to increased mitochondrial production of semiquinone and, consequently, mitochondrial superoxide; (ii) specific redox-cycling of unprenylated quinones and ascorbate mainly in the impaired cancerous mitochondria due to their redox imbalance; and (iii) downregulation of tumor-associated NADH oxidase (tNOX). These processes cause severe oxidative stress in cancerous mitochondria and accelerated depletion of ATP selectively in cancer cells, which are detrimental to them. However, this strategy needs further development and improvement to be successfully and efficiently applied in vivo and transferred to the clinic as an adjuvant to conventional anticancer therapy to increase its effectiveness and tolerability.

## 4. Materials and Methods

### 4.1. Chemicals

L-Ascorbic acid and menadione (K3) were purchased from Sigma-Aldrich (Weinheim, Germany). Quinones were purchased from the following companies: 2-isopropyl-5-methyl-1,4-benzoquinone (thymoquinone, ThymoQ) and atovaquone (AtovaQ)—Tokyo Chemical Industry (TCI, Tokyo, Japan); 2,6-dimethoxy-1,4-benzoquinone (DMBQ), menadione, and 2-bromo-1,4-napthoquinone—Sigma-Aldrich (St. Louis, MO, USA); 2-methoxy-5-methylcyclohexa-2,5-diene-1,4-diene (MMBQ)—Thermo Fisher Scientifics (ACROS Organics, India); coenzyme Q0 (CoQ0), coenzyme Q1 (CoQ1), and coenzyme Q10 (CoQ10), menaquinone (K2), and phylloquinone (K1)—Cayman (Ann Arbor, MI, USA).

All reagents used in the experiments were “analytical grade” or “HPLC-grade”.

### 4.2. Cells and Treatment Protocol

Leukemic lymphocytes (Jurkat; RIKEN Bioresource Center, Saitama, Japan) derived from patients with acute lymphoblastic leukemia, as well as normal lymphocytes derived from clinically healthy blood donors (Human Peripheral Blood Cells; Cell Applications Inc., San Diego, CA, USA) were cultured in RPMI-1640 medium (Sigma-Aldrich, Weinheim, Germany), supplemented with 10% heat-inactivated fetal bovine serum (FBS; Gibson, Nashville, TN, USA) and antibiotics (100 U/mL penicillin and 100 μg/mL streptomycin) (Gibson, USA) in a humidified atmosphere at 37 °C, saturated with 5% CO_2_. The cells were collected by centrifugation (1000× *g*, 10 min for leukemia lymphocytes and 1500× *g*, 15 min for normal lymphocytes) and placed in a fresh medium without antibiotics prior to treatment.

Experiments were also performed on adhesive cells lines: (i) colon epithelial cells—cancer (Colon26) and normal (FHC); (ii) breast epithelial cells—cancer (MFC7) and normal (MCF10A); (iii) glioblastoma cells (U87MG, GS9L) and normal microglial cells (EOC2). All cell lines were purchased from the American Type Culture Collection (ATCC, USA). FHC and MCF10A were cultured in DMEM-F12 (Sigma-Aldrich, Weinheim, Germany) and DMEM (Sigma-Aldrich, Weinheim, Germany), respectively, both supplemented with growth factors. EOC2 were cultured in LADMAC conditioned DMEM. Colon26, MCF7, U87MG, and GS9L were cultured in DMEM supplemented with antibiotics. All mediums were supplemented with 10% FBS. Cells were cultured in a humidified atmosphere at 37 °C, saturated with 5% CO_2_. Twenty-four hours before the treatment, the cells were replaced in a fresh medium without growth factors. To remove the adhesive cells from the plates, we used a trypsin-EDTA solution (0.5% of trypsin, 0.2% of EDTA) or cell scraper and subsequent washing with phosphate-buffered saline (PBS). During the culturing and experiments, the adhesive cells were sedimented by centrifugation (800× *g*/5 min). The concentration of glucose in the cell cultured medium was standard (2 mM).

Ascorbate was dissolved in PBS (10 mM, pH 7.4). Quinones were dissolved in DMSO to 50 mM stock solution and then several working solutions in PBS were prepared. The final concentration of DMSO in the cell suspension was below 1%. At this concentration, DMSO did not influence cell viability.

### 4.3. Cell Viability and Proliferation Assay

Cell viability and proliferation were analyzed by trypan blue staining and automated counting, using Countess™ Automated Cell Counter (Invitrogen, Eugene, OR, USA).

Briefly, 10 μL of trypan blue (0.4%) was added to 10 μL of cell suspension, incubated for 30 s, and 10 μL of the cell suspension was placed in a Countess^®^ (Invitrogen) glass chamber. The number of live and dead cells in the suspension was counted automatically. The linear range to operate with the automated cell counter was 1 × 10^4^–5 × 10^6^ cells/mL, and the optimal cell size was in the range of 5–60 μm.

### 4.4. Mitochondrial Superoxide Assay

MitoSOX™ Red Mitochondrial Superoxide Indicator (Molecular Probes, Invitrogen) was used to analyze superoxide in live cells. Once in the mitochondria, MitoSOX™ Red reagent is oxidized by superoxide and exhibits red fluorescence. The probe is not oxidized by other ROS/RNS, and its oxidation is prevented by superoxide dismutase [80].

Briefly, MitoSOX™ Red was dissolved in DMSO to 5 mM stock solution, which was diluted with Hank’s Balanced Salt Solution (HBSS, containing Ca^2+^ and Mg^2+^) to prepare 3 μM MitoSOX™ Red working solution on the day of experiment. Cell cultured medium was removed and 1 mL of 3 μM MitoSOX™ Red solution was added to 6-well plates containing cells (5 × 10^5^ cells per well). The cells were incubated for 30 min at 37 °C, protected from light, washed three times with PBS, and finally collected and re-suspended in 1 mL of PBS. The fluorescence intensity was detected immediately at λ_ex_ = 510 nm and λ_em_ = 580 nm, using a microplate reader (TECAN Infinite^®^ M1000, Grödig, Austria). Values were normalized to equal number of cells in samples.

### 4.5. ATP Assay

Steady-state ATP levels in the cells were analyzed by CellTiter-Glo^TM^ Luminescent Cell Viability Assay (Promega, Madison, WI, USA), based on quantification of ATP from the luminescence signal generated by the luciferase-catalyzed conversion of luciferin to oxyluciferin in the presence of ATP, Mg^2+^ and molecular oxygen.

Briefly, 100 μL aliquots of cell suspensions were placed in 96-well plates and incubated with 100 μL of CellTiter-Glo^TM^ reagent containing luciferin and luciferase, using the protocol recommended by the manufacturer. The luminescence, produced by the luciferase-catalyzed conversion of luciferin into oxyluciferin in the presence of ATP produced by viable cells, was detected using a microplate reader (TECAN Infinite^®^ M1000), working in a chemiluminescent mode. The linear range for this assay was up to 5 × 10^5^ cells per well. Values were normalized to equal number of cells in samples.

### 4.6. Total Antioxidant Capacity (TAC) Assay

The TAC assay was performed on tissue lysates using OxiSelect^TM^ Total Antioxidant Capacity (TAC) Assay kit (Cell Biolabs, Inc., San Diego, CA, USA). The method is based on the reduction of Cu^2+^ to Cu^+^ by antioxidants and other reducing equivalents in the biological sample. Cu^+^ interacts with a chromophore to obtain a color product with an absorption maximum at 490 nm. The value of the absorption is proportional to the total antioxidant, respectively, reducing capacity of the biological object.

Briefly, tissue lysates were prepared as it was described in the manufacturer’s instruction. All lysates were adjusted to the same protein concentration and 20 μL of aliquots were placed in a 96-well plate. Each sample was incubated with cupper ion reagent and chromophore as it was described in the instruction. The absorption of the product at 490 nm was detected by a microplate reader (Tecan Infinite F200 PRO). Three independent experiments were performed for each homogenate, with two parallel sample measurements for each experiment.

The total antioxidant capacity of the samples was determined by a calibration curve using uric acid as a standard. The results are presented as a “Total Antioxidant Capacity (TAC)”, which is equivalent to “Total Reducing Capacity” (TRC) in “mM Uric Acid Equivalents”. One mM of uric acid corresponds to 2189 μM of Cu^2+^-reducing equivalents.

### 4.7. tNOX (ENOX2) Assay

Ecto-NOX disulfide-thiol exchanger 2 (ENOX2, tNOX) expression was detected in tissue lysates using Mouse ENOX2 ELISA kit (LifeSpan BioScience, Seattle, WA, USA). Tissue lysates were prepared and analyzed as it was described in the manufacturer’s instruction. The protein concentration in the lysates was determined by Bradford assay. tNOX analysis is based on the quantitative sandwich enzyme immunoassay technique. The antigen-antibody complex was detected spectrophotometrically at 450 nm, based on the oxidation of 3,3′,5,5′-tetramethylbenzidine (TMB) by horseradish peroxidase conjugated to avidin, which interacts with a biotinylated secondary antibody in the sandwich. Lyophilized tNOX (ENOX2) protein was used as a standard. All samples were run in triplicate.

### 4.8. Animals and Treatment Protocols

The animal experiments in this study were approved by the National Institutes for Quantum Science and Technology (QST) Institutional Animal Care and Use Committee (protocol #14-1006), Chiba, Japan, and all experiments were performed in accordance with relevant guidelines and regulations.

BALB/c nude mice were obtained from Charles River Labs (Japan). All mice were male and were used at 6–8 weeks of age and maintained in specific pathogen-free conditions. The mice were placed on a vitamin C and K3 deficient diet (CLEA, Tokyo, Japan) one day before cell transplantation. The diet contained the minimum amount of vitamin K1, which is essential for mice. Since we are investigating the effects of Q/A redox-pairs, and K3/A is one of them, the exclusion of vitamins K3 and C from the diet aimed to eliminate the effect of oral administration of K3/A combination on that of parenteral administration of CoQ0/A and K3/A.

Glioblastoma model (hind paw xenografts): The experimental design is shown in Appendix A. U87MG cells (1 × 10^6^ cells in 50 mL) were inoculated subcutaneously into the hind paw of the mouse to trigger a development of glioblastoma. The glioblastoma-bearing mice were divided into the following groups: (i) control group—single s.d. injection of saline solution near the tumor; (ii) 1× CoQ0/A-treated group—single s.d. injection of CoQ0/A (70 μg/7 mg per kg body weight) near the tumor; (iii) 2× CoQ0/A-treated group—two s.d. injections of CoQ0/A (70 μg/7 mg per kg body weight) near the tumor with three days interval between injections. The volume of each injection was 50 μL.

Colon cancer model (hind row xenografts): The experimental design is shown in Appendix A. Colon26 cells (1 × 10^6^ cells in 50 µL) were inoculated subcutaneously into the hind paw of the mouse to trigger the development of colon cancer. The colon-cancer-bearing mice were divided into the following groups: (i) control group—six s.d. injections of saline solution near the tumor—twice per week; (ii) 6× CoQ0/A-treated group—six s.d. injections of CoQ0/A (70 μg/7 mg per kg body weight) near the tumor—twice per week; (iii) 6× K3/A-treated group—six s.d. injections of K3/A (70 μg/7 mg per kg body weight) near the tumor—twice per week. The volume of each injection was 50 μL.

Body weight was measured once or twice per week. The duration of the experiments was 45 days in the case of glioblastoma model and 80 days in the case of colon cancer model. The approved humane endpoint was three months after cell transplantation. Mice were subjected to euthanasia by using pentobarbital (Somnopentyl, Kyoritsu Seiyaku, Co., Tokyo, Japan; 150 mg per kg b.w.) through cervical dislocation under 4% isoflurane anesthesia. However, the mice were also euthanized at the following conditions: when the tumor size exceeded 1500 mm^3^, at rapid weight loss of 25%, headedness and/or tetraplegia.

### 4.9. In Vivo MRI Measurements

Each mouse was anesthetized with isoflurane (3% for initial induction and 1–2% during MRI scanning) and was placed in the prone position on a custom-built MRI stage with a bite bar and a facemask. This is the only anesthesia used in the study. The respiration rate was monitored using a respiration sensor (SA Instruments, Inc., Stony Brook, NY, USA) and was regulated at 80–120 breaths per minute. The core body temperature was monitored with a rectal probe (FOT-L and FTI-10, FISO Technologies Inc., Germany) and was regulated at 37.0 ± 1.0 °C using a water circulating pad and a warm circulation air system. MRI data were acquired using a horizontal 7.0-T Bruker BioSpec 70/40 MRI system with an 86 mm volume transmit and a 4-channel phased array receiving cryoprobe (Bruker Biospin, Ettlingen, Germany). The software and console of the MRI scanner was ParaVision 360 and AVANCE NEO, respectively. Following the standard adjustment routines, pilot scans (Triplet sequence) were used for accurate positioning of the animal head inside the magnet.

The T_2_W images were obtained using a spin-echo 2D-RARE (rapid acquisition with relaxation enhancement) pulse sequence with the following parameters: repetition time = 3000 ms, effective echo time = 60 ms, RARE factor = 8, field of view = 16 × 16 mm^2^, matrix size = 160 × 160, in-plane resolution = 0.1 × 0.1 mm^2^, number of slices = 13, slice thickness = 0.3 mm, slice gap = 0 mm, fat suppression = on, and number of averages = 8.

### 4.10. Statistical Analysis

All results are expressed as means ± standard deviation (SD). The normality of the distribution for all parameters of each experimental group in vivo was initially confirmed by using the Kolmogorov–Smirnov test. The most extreme differences for all experimental groups were below the critical D-values. Based on the normality of distribution in all groups, the comparisons between them were performed using Student’s *t*-test for multiple comparisons. Two-tailed *p* values of less than 0.05 were considered statistically significant.

## Figures and Tables

**Figure 1 ijms-24-08435-f001:**
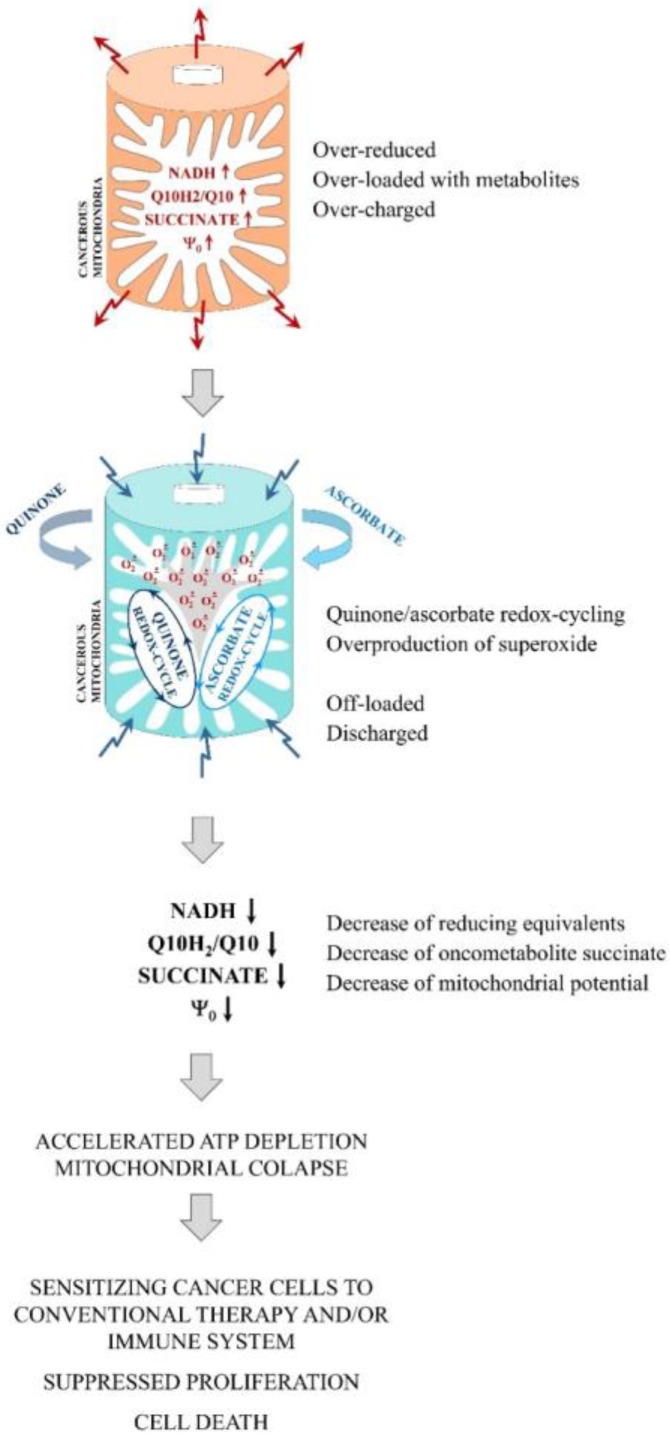
Molecular mechanism for targeting and impairing mitochondrial function by quinone and ascorbate redox-cycling in cancer cells (adapted from Bakalova et al. [9]).

**Figure 2 ijms-24-08435-f002:**
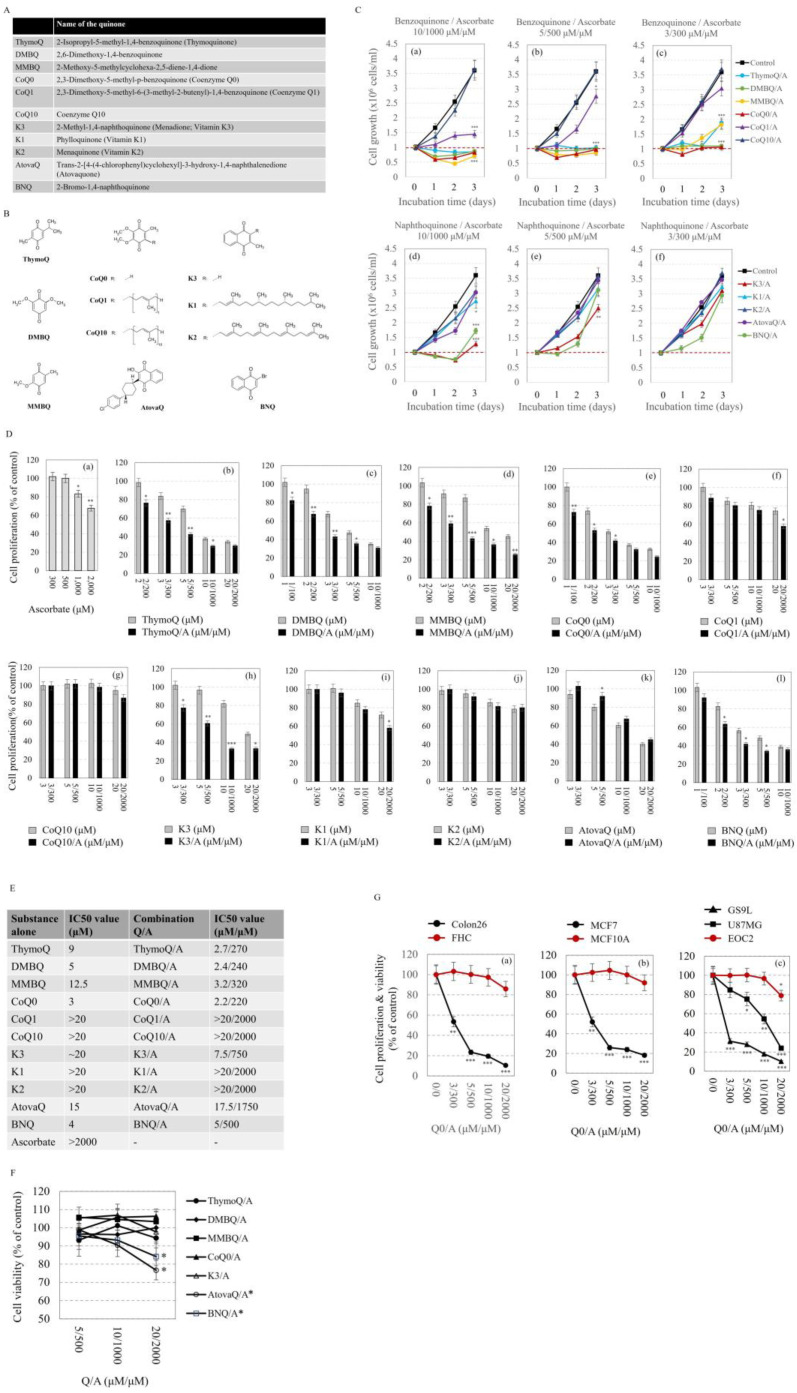
Effects of quinone/ascorbate (Q/A) redox-pairs on cancer cell proliferation and viability—comparison with normal cells of the same origin. (**A**,**B**) The names and chemical formulas of the investigated quinones. (**C**) Kinetic curves of proliferation of Q/A-treated cancer cells. Untreated cells were used as control. Incubation conditions: Leukemia cells (Jurkat; 1 × 10^6^ cells/mL) were incubated with the respective Q/A combination at different concentrations within three days. * *p* < 0.05, ** *p* < 0.01, *** *p* < 0.001 versus control (untreated) cells at three days incubation. The red dashed lines indicate the initial number of live cells in the samples. (**D**) Effects of ascorbate and respective quinones administered alone and in Q/A combination on cancer cell proliferation. Untreated cells were used as controls and cell proliferation in these samples was considered 100%. Incubation conditions: Leukemia cells (Jurkat; 1 × 10^6^ cells/mL) were incubated with the respective quinone or Q/A combination at different concentrations within 48 h. * *p* < 0.05, ** *p* < 0.01, *** *p* < 0.001 versus quinone-treated cells at the respective concentration or versus untreated cells in the case of cells treated with ascorbate only (a). (**E**) IC50 values calculated for quinone-treated and Q/A-treated Jurkat cells within 48 h. (**F**) Effects of selected Q/A combinations on viability of normal lymphocytes. Incubation conditions: Normal lymphocytes (1 × 10^6^ cells/mL) were incubated with the respective Q/A combination at different concentrations within 48 h. * *p* < 0.05 versus untreated normal lymphocytes at the respective concentration. (**G**) Effects of coenzyme Q0/ascorbate (CoQ0/A) redox-pair on proliferation and viability of different cell lines: (a) colon cancer (Colon26) and normal colon epithelial (FHC); (b) breast cancer (MCF7) and normal breast epithelial (MCF10A); (c) glioblastoma (U87MG, GS9L) and normal microglial cells (EOC2). Incubation conditions: Cells (5 × 10^5^ cells/mL) were incubated with Q0/A at different concentrations within 48 h. * *p* < 0.05, ** *p* < 0.01, *** *p* < 0.001 versus untreated cells at the respective concentration. In all charts, data are presented as means ± SD from three independent experiments with two parallel measurements per sample.

**Figure 3 ijms-24-08435-f003:**
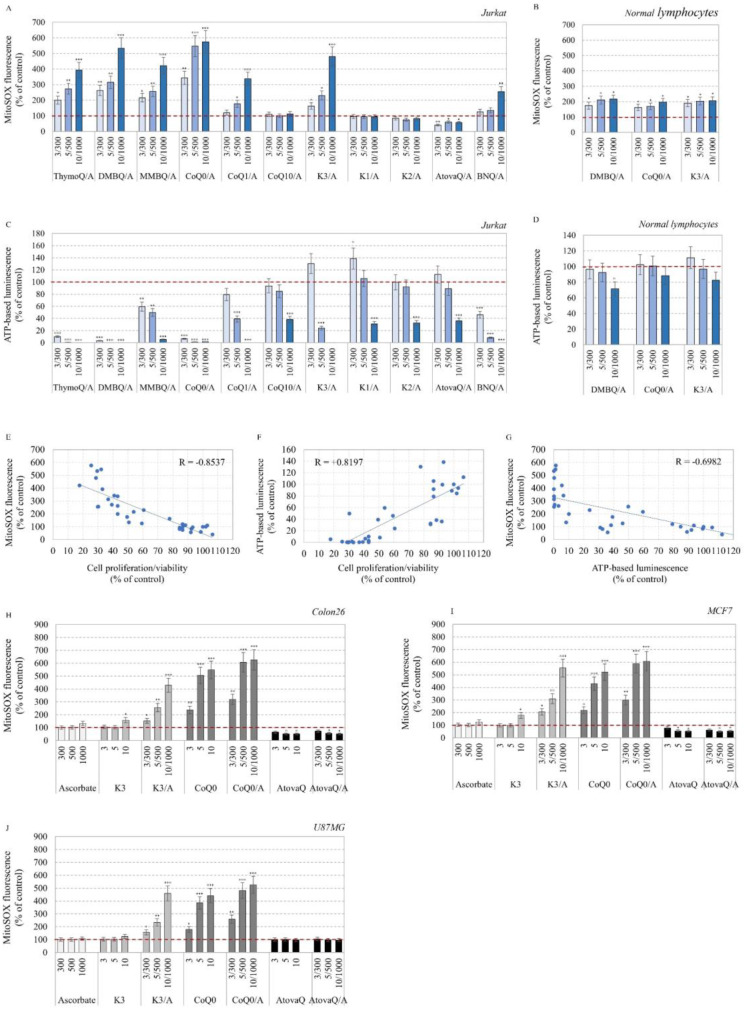
Effects of quinone/ascorbate (Q/A) redox-pairs on mitochondrial function of cancer and normal cells of the same origin. (**A**) Mitochondrial superoxide level in Q/A-treated leukemia lymphocytes (Jurkat) within 48 h analyzed by MitoSOX fluorescence. Untreated cells were used as control and MitoSOX fluorescence in these samples was considered as 100% (red dashed line). (**B**) Mitochondrial superoxide level in Q/A-treated normal lymphocytes within 48 h analyzed and calculated as in (**A**). (**C**) Steady-state ATP level in Q/A-treated leukemia lymphocytes (Jurkat) within 48 h analyzed by CellTiterGlo^TM^ luminescence. Untreated cells were used as control and ATP-based luminescence in these samples was considered as 100% (red dashed line). (**D**) Steady-state ATP level in Q/A-treated normal lymphocytes within 48 h analyzed and calculated as in (**C**). (**E**–**G**) Correlation analysis between mitochondrial superoxide, ATP level, and cell proliferation/viability in Q/A-treated leukemic lymphocytes, presented in Figure 2D and Figure 3A,C. R—correlation coefficient. (**H**–**J**) Mitochondrial superoxide level in cancer cells (Colon26, MCF7, U87MG) treated with ascorbate, CoQ0/A, K3, and AtovaQ administered alone or in combination within 48 h and analyzed by MitoSOX fluorescence. Untreated cells were used as controls and MitoSOX fluorescence in these samples was considered as 100% (red dashed line). In all charts, data are presented as means ± SD from three independent experiments with two parallel measurements per sample. All values were normalized to equal number of cells in samples. * *p* < 0.05, ** *p* < 0.01, *** *p* < 0.001 versus control (untreated) cells.

**Figure 4 ijms-24-08435-f004:**
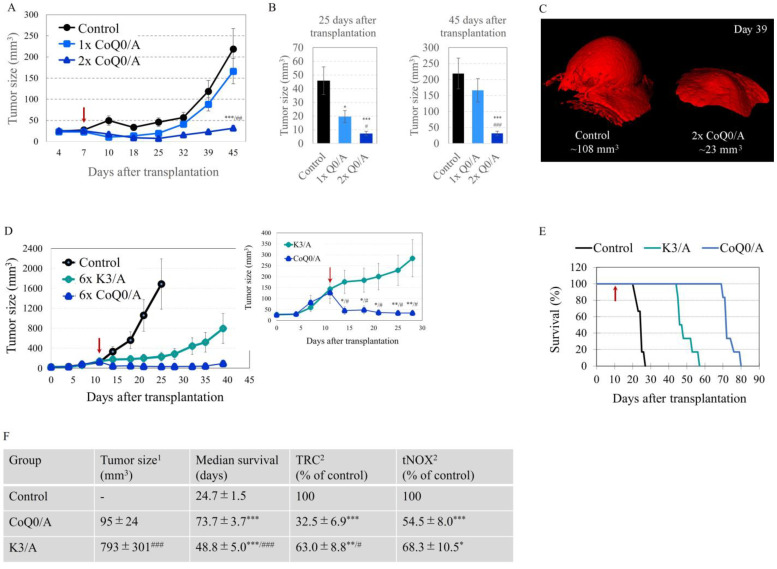
Effects of coenzyme CoQ0/ascorbate (CoQ0/A) and menadione/ascorbate (K3/A) on cancer-bearing mice—hind paw xenografts: (**A**) Effect of subdermal (s.d.) injection of CoQ0/A on tumor growth in glioblastoma-grafted mice, detected within 45 days after cell (U87MG) transplantation: Control—single injection of saline solution (n = 3); 1× CoQ0/A—single injection of CoQ0/A on day 7th after cell transplantation (n = 3); 2× CoQ0/A—two injections of CoQ0/A on days 7th and 10th after cell transplantation (n = 3). CoQ0/A was injected near the tumor in a single dose 70 μg/7 mg per kg body weight (50 μL volume). Data are presented as means ± SD from 3 mice at each time point. *** *p* < 0.001 versus control group; ^##^
*p* < 0.01 versus 1× CoQ0/A-treated group. (**B**) Comparison of tumor size between control group and CoQ0/A-treated group, measured 25 days and 45 days as shown in (**A**). * *p* < 0.05, *** *p* < 0.001 versus control group; ^#^
*p* < 0.05, ^###^
*p* < 0.001 versus 1× CoQ0/A treated group. (**C**) Representative 3D magnetic resonance images of tumors in a control mouse and 2× CoQ0/A-treated mouse obtained 39 days after glioblastoma cell transplantation. (**D**) Effect of s.d. injection of CoQ0/A and K3/A on tumor growth in colon cancer-grafted mice, detected within 39 days after cell (Colon26) transplantation: Control—6 s.d. injections of saline solution (twice per week) (n = 6); 6× CoQ0/A—six s.d. injections of CoQ0/A (twice per week) (n = 6); 6× K3/A—six s.d. injections of K3/A (twice per week) (n = 6). CoQ0/A and K3/A were injected near the tumor in a single dose 70 μg/7 mg per kg body weight (50 μL volume), starting from day 11 after cell transplantation (red arrow). Data are presented as means ± SD from 6 mice at each time point. * *p* < 0.05, ** *p* < 0.01 versus K3/A treated group; ^#^
*p* < 0.05 versus the initial tumor size (red arrow). (**E**) Effect of s.d. injection of CoQ0/A and K3/A on the survival of mice described in (**F**). Effects of 6× CoQ0/A and 6× K3/A on tumor growth, median survival, tissue reducing capacity and tNOX expression in the tumors of colon cancer grafted mice—a comparative analysis. Data are presented as means ± SD from six mice in each group for tumor size and median survival and three mice in each group with three measurements for each specimen for TRC and tNOX assays. TRC and tNOX were analyzed on day 22 after transplantation. Samples isolated from untreated mice were used as controls. Data are expressed as % of the respective control. * *p* < 0.05, ** *p* < 0.01, *** *p* < 0.001 versus untreated (control) mice; ^#^
*p* < 0.05, ^###^
*p* < 0.001 versus CoQ0/A-treated mice.

**Figure 5 ijms-24-08435-f005:**
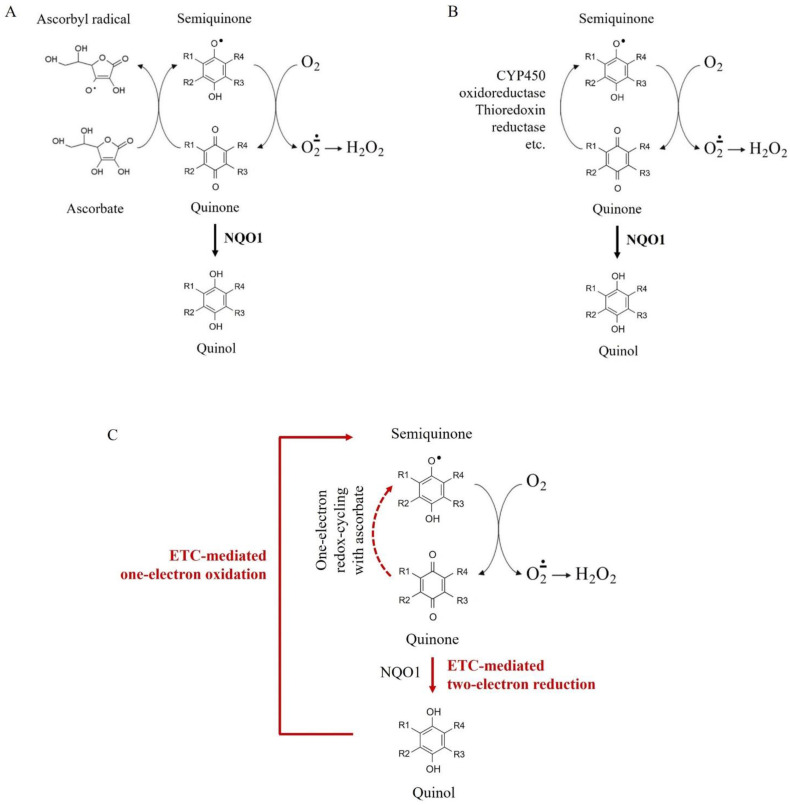
Schematic representation of redox-cycling mechanisms of quinones with production of superoxide and hydrogen peroxide: (**A**) non-enzymatic ascorbate-driven one-electron redox-cycling; (**B**) enzyme-facilitated one-electron redox-cycling catalyzed by cytochrome P450 oxidoreductase (CYP450), thioredoxin reductase. etc.; (**C**) Redox-cycling mechanisms of quinone mediated by the electron-transport chain (ETC) of impaired cancerous mitochondria. Note: NAD(P)H:quinone oxidoreductase 1 (NQO1) maintains quinone in its reduced (enol) form. Thus, NQO1 restricts and even prevents non-enzymatic ascorbate-driven and enzyme-mediated one-electron redox-cycling of quinone to semiquinone and production of superoxide. One-electron oxidation of quinol in the mitochondrial ETC restores semiquinone and the subsequent generation of mitochondrial superoxide, as well as local non-enzymatic ascorbate-driven one-electron redox-cycling in the mitochondria.

## Data Availability

Data are available by request on the following e-mail address: bakalova.rumiana@qst.go.jp; bakalovazheleva@gmail.com.

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
