# Peer review of "Redox-Cycling “Mitocans” as Effective New Developments in Anticancer Therapy"

_ijms, 2023, doi:10.3390/ijms24098435_

Round 1
Reviewer 1 Report
The cancer and its treatment is one of the “hot topics” in current research strategies in many countries. New therapeutic solutions for cancer diseases certainly not only increase the level of our knowledge about the pathomechanism of cancer but also propose new, so far unknown therapeutic strategies.
The presented manuscript proposes using several chemical compounds to disrupt the normal function of the mitochondria of cancer cells. Nevertheless, I expressed considerable concern that the proposed therapy should be tested as part of the supplement/support of classical treatment methods rather than constitute an alternative therapy pathway.
The authors have prepared a very extensive description of the subject in the introduction to the manuscript. Unfortunately, therefore, there is no clearly outlined purpose of the manuscript mentioned above. Consequently, the readers may feel lost when dealing with so much information.
In the methodology, the authors also made minor editorial errors. For example, when describing an animal model, the specification of the number of tumor cells that were administered to the animals, their volume, and which time points should be highlighted. In addition, it is interesting to note that vitamins K and C in the diet of animals are limited; why did the authors use that method? This requires careful argumentation.
Evaluation of lymphocytes after 5 days without prior stimulation is unreliable. Did the authors use specific stimulants for lymphocytes, e.g., PHA, in their research?
Did the authors confirm the expression of the NRF2 and KEAP1 genes in the cancer lines used? These genes are one of the critical signaling pathways responsible for maintaining the correct redox balance in the body.
How long did the authors run the animal experiment? Whether a single administration of chemical compounds was enough to obtain a long-term effect or whether a series of injections had to be performed.
Did the authors also assess the status of cancer-related cells, i.e., TAMs, CAFs, etc.? How did the supply of these compounds affect the tumor microenvironment?
Author Response
General response
The manuscript was substantially revised according to the Reviewers' comments.
Figures 2, 3, and 4 were revised and replaced in the R1 version. Revised figures contain symbols for statistical significance between samples.
A new table was added to the Supporting information to illustrate the synergistic antiproliferative and cytotoxic effect of quinone and ascorbate in the investigated Q/A redox-pairs (please, see Table S1).
Some errors in the description of the experimental animal protocol that we noticed have also been corrected.
The manuscript was edited by our native English-speaking colleague.
All corrections are mentioned in red into the Main text and Supporting information files.
Point-by-point response to the Reviewers is given below.
We would like to thank the reviewers for their constructive comments and suggestions that improve the quality of our manuscript!
POINT-BY-POINT RESPONSE
Response to Reviewer #1
- Reviewer's Comment: I expressed considerable concern that the proposed therapy should be tested as part of the supplement/support of classical treatment methods rather than constitute an alternative therapy pathway.
Answer: We agree with the reviewer that it is overstated at this stage to claim that Q/A combination drugs can be an alternative to conventional anticancer therapy. However, experimental data show that they have the potential to support and improve the effectiveness of conventional therapeutic strategies in cancer. This point of view was briefly described at the end of the Conclusion. Please, see page 18 of the revised version (lines 634-636):
“However, this strategy needs further development and improvement to be successfully and efficiently applied in vivo and transferred to the clinic as an adjuvant to conventional anticancer therapy to increase its effectiveness and tolerability.”
- Reviewer's Comment: There is no clearly outlined purpose of the manuscript.
Answer: At the end of the introduction, we redefined the main goals of our study based on the outstanding (unclarified) questions of the topic. Please, see page 4 of the revised version (lines 138-147):
“The aim of the present study was: (i) to elucidate the effectiveness of Q/A redox-pairs (containing CoQ and K3 analogues) as anticancer combination drugs and potential tolerance to normal cells and tissues as a result of their redox-cycling in impaired mitochondria; (ii) to compare the effects of unprenylated versus prenylated quinones and to assess the impact of prenylation on selective Q/A-mediated anticancer effects; (iii) to compare the effects of benzoquinone/ascorbate versus naphthoquinone/ascorbate redox-pairs in vitro and in vivo, considering which of the two combinations is more efficient, tolerant, and promising for translational research in the future. Eleven Q/A redox-pairs were investigated, and the experiments were performed under identical conditions on cultured cells and cancer-bearing mice.”
- Reviewer's Comment: In the methodology, the authors also made minor editorial errors. For example, when describing an animal model, the specification of the number of tumor cells that were administered to the animals, their volume, and which time points should be highlighted. Answer: The number of tumor cells that were inoculated in the animals and their volume were given in Figure S1 in the Supporting information. Time points of drug administration (after cell transplantation) were highlighted with red arrows in Figure 4. Please, see revised Figure S1 in the Supporting information and Figure 4 in the Main text.
- Reviewer's Comment: In the methodology, it is interesting to note that vitamins K and C in the diet of animals are limited; why did the authors use that method? This requires careful argumentation.
Answer: A brief explanation of the exclusion of pro-vitamin K3 (menadione) and vitamin C (ascorbate) from the diet is given in the Methods. Please, see page 20 of the revised version (lines 748-752):
“The diet contained the minimum amount of vitamin K1, which is essential for mice. Since we are investigating the effects of Q/A redox-pairs and K3/A is one of them, the exclusion of vitamins K3 (menadione) and C (ascorbate) from the diet aimed to eliminate the effect of oral administration of K3/A combination on that of parenteral administration of CoQ0/A and K3/A.”
We also found other inaccuracies in the description of the two models, which have been corrected. Please, see page 21 of the revised version (lines 756-773).
- Reviewer's Comment: Evaluation of lymphocytes after 5 days without prior stimulation is unreliable. Did the authors use specific stimulants for lymphocytes, e.g., PHA, in their research?
Answer: We agree with the reviewer that it is not correct and reliable to assess the viability and proliferation of leukemic lymphocytes after 3 days due to depletion of glucose and glutamate in the medium. We did not use any specific stimulant in this experiment. Therefore, we corrected Figure 2C, presenting the kinetic curves up to 3 days of incubation. Please, see Figure 2C in the revised version.
- Reviewer's Comment: Did the authors confirm the expression of the NRF2 and KEAP1 genes in the cancer lines used? These genes are one of the critical signaling pathways responsible for maintaining the correct redox balance in the body.
Answer: Unfortunately, we cannot provide data on the expression of NRF2 and KEAP1 genes in our study. We found no articles reporting the effect of Q/A redox-pairs on these two factors. NRF2 is definitely within the scope of our interests and future research.
- Reviewer's Comment: How long did the authors run the animal experiment?
Answer: The following sentence was included in the Methods (please, see page 21 of the revised version, lines 771-773):
“The duration of the experiments was 45 days in the case of glioblastoma model and 80 days in the case of colon cancer model.”
- Reviewer's Comment: Whether a single administration of chemical compounds was enough to obtain a long-term effect or whether a series of injections had to be performed.
Answer: A single administration of compounds was not enough to obtain a long-term effect. We compared single injection of CoQ0/A with two injections (twice per week) of CoQ0/A on glioblastoma model, as well as six injections (twice per week) of CoQ0/A or K3/A on colon cancer model. It was described in the Methods, as well as in the Legend of Figure 4. Please, see page 12 (legend to Figure 4, (lines 347-351 and 357-361) and page 21 of the revised version (lines 753-770).
- Reviewer's Comment: Did the authors also assess the status of cancer-related cells, i.e., TAMs, CAFs, etc.? How did the supply of these compounds affect the tumor microenvironment?
Answer: In this study, we did not analyze the effects of Q/A redox-pairs on the status of tumor-associated non-cancerous cells. However, in other study on intracranial (orthotopic) glioblastoma model in mice we analyzed the effects of K3/A and CoQ0/A on tumor cell density and tumor blood flow, using native MRI on anesthetized animals. These data provide indirect information about the effect of the two combinations on the tumor microenvironment. The data obtained for K3/A have been published recently (ref. 11 cited in the revised version), and those for CoQ0/A will be published elsewhere. We included a brief explanation of these data in the end of Discussion. Please, see page 18 of the revised version (lines 604-616):
“It has been found that Q/A redox-pairs may influence the integrity of the tumor microenvironment and the activity of tumor-associated immune cells and fibroblasts. We recently reported that K3/A treatment significantly decreased tumor cell density and increased tumor perfusion, which is indirect evidence of its influence on the tumor microenvironment [11]. Similar effects were observed in CoQ0/A-treated glioblastoma mice - intracranial model (data will be published elsewhere). These findings were consistent with published histological data obtained on K3/A-treated cancer cells and tissues [83]. K3/A (at certain doses) has been found to cause a specific form of cell death called autoschizis, characterized by a reduction in cell size due to loss of cytoplasm by self-excision without loss of cell organelles, morphological degradation of the nucleus, and formation of apoptotic bodies [83]. CoQ0 and K3 have also been found to affect macrophage activity and inhibit inflammation by targeting the NLRP3 inflammasome [84, 85]. K3/A suppressed PD-L1 expression in cancer cells [9].”

Reviewer 2 Report
The research article entitled, “Redox-cycling “mitocans” as effective new developments in anticancer therapy”, by Rumaiana et al., proposes a pharmacological strategy targeting cancer cell mitochondria with redox-cycling drugs. They used different pairs of Q/As and tested their effect on various parameters such as cell proliferation, mitochondrial superoxide production, ATP level, etc. Also, they confirmed their findings on the tumor xenograft model on mice. Most of the data are adequately represented to support the authors’ hypothesis. However, there is room for improvement before being considered for publication.
Major points
1. Statistical analysis of significance (p-value) are missing in Figure 2C, D, G. This makes the reviewer difficult to agree to the authors’ claim in the result section.
Line 164: The antiproliferative effects of benzoquinone/ascorbate combinations were significantly stronger than that of naphthoquinone/ascorbate combinations
Line 185 : ThymoQ/A, DMBQ/A, MMBQ/A, CoQ0/A, K3/A, and BNQ manifested synergistic antiproliferative and cytotoxic effects. -> Cannot conclude without p-value calculation
Line 191 : The antiproliferative effects of CoQ1/A and K1/A were additive, whereas the effects of CoQ10/A and K2/A were rather antagonistic.
Line 200 : AtovaQ/A and 200 BNQ/A demonstrated relatively weak but significant cytotoxicity to normal lymphocytes at a concentration of 20/2000 μM/μM (Figure 2F).
Line 269-273 : The effects were synergistic, with synergism being more pronounced with K3/A than with CoQ0/A. AtovaQ/A significantly decreased the level of mitochondrial superoxide in Colon26 and MCF7 but its effect in U87MG cells was negligible (Figure 3H). No synergism was found between quinone and ascorbate
Line 496: However, no synergism was observed between AtovaQ and ascorbate
->These cannot be concluded without proper p-value calculation and labeling on the figure.
2. Like in Figure 2, p-value calculation is completely missing in Figure 3.
3. Line 236: effect was moderate. -> Define moderate. It seems like the effect is smaller than others, but still there. Again, difficult to tell without proper statistical calculations.
4. Line 253-258 -> Specify which condition was used for calculation.
5. Line 373: while K3/A caused only a growth arrest -> where did this conclusion came from? 4D? Also, p-value calculation needed.
6. Indication of the number of technical/biological replicas in the figure legends is missing.
Minor points
1. Line 83: adapted by…-> adapted from
2. Line 127: Study-state->Steady-state
3. Line 146-147: Their names and chemical formulas are given in Figure 2A, B. They are coenzyme Q and K3 analogues. -> Their names and chemical formulas are given in Figure 2A and B, which are coenzyme Q and….
4. Line157: DMBQ, MMBQ->DMBQ and MMBQ
5. Line 166: Figure 2C-1,b,c->Should be Figure 2C-a,b,c
6. Line 174: 20/2000 uM/uM->cannot find 10/2000 in figure 2C-a,b. Please check.
7. Line 188: Figure 2D-I->Figure 2D-i
8. Figure 2A->Difficult to read because of low resolution.
9. Line 232, 243->Please add some transitional phrase such as, In order to… Nest, we wanted to…, etc.
10. Line 337: @g/7 mg per kg body weight->Unit correction needed.
11. Line 373: while K3/A caused only a growth arrest->Where did this conclusion come from? 4D? Please specify, with statistical analysis.
12. Lilne 398: Figure 2C, F->Should be 2G and F. Please check.
13. Line 422: such a mechanism is unlikely to prenylated quinones…->such a mechanism is unlikely to (affect?) prenylated… Please check the expression.
14. Line 430: overchanged->overcharged?
15. Line 490: Figure 2A, H, I, J->Should be Figure 3A, H, I, and J
16. Line 496: no synergism was observed between AtovaQ and ascorbate…->Again, without statistics, you cannot conclude your interpretation.
17. Line 539: in vitro and in vivo are italicized, but some of the others throughout the manuscript (e.g., Line 86, 142, 327…) were not. Correction needed.
18. Line 594: mail goals->main goals, perhaps?
19. Line 619: as well as on normal…-> Delete ‘on’.
20. Line 653: 5-60@m->Unit check. 5-60 micrometer, perhaps?
21. Line 707. Animal and treatment protocols->Details of the animal numbers used are missing.
Author Response
General response
The manuscript was substantially revised according to the Reviewers' comments.
Figures 2, 3, and 4 were revised and replaced in the R1 version. Revised figures contain symbols for statistical significance between samples.
A new table was added to the Supporting information to illustrate the synergistic antiproliferative and cytotoxic effect of quinone and ascorbate in the investigated Q/A redox-pairs (please, see Table S1).
Some errors in the description of the experimental animal protocol that we noticed have also been corrected.
The manuscript was edited by our native English-speaking colleague.
All corrections are mentioned in red into the Main text and Supporting information files.
Point-by-point response to the Reviewers is given below.
We would like to thank the reviewers for their constructive comments and suggestions that improve the quality of our manuscript!
POINT-BY-POINT RESPONSE
Response to Reviewer #2
Major points
- Reviewer's Comment: Statistical analysis of significance (p-value) are missing in Figure 2C, D, G. This makes the reviewer difficult to agree to the authors’ claim in the result section.
Answer: P-values were included in the charts of Figure 2C, D, F, G. A new table was included in the Supporting information for easier understanding of the synergistic antiproliferative effect of quinone and ascorbate in leukemic lymphocytes. Please, see the revised Figure 2 and Table S1 in the Supporting information.
- Comment on line 164: “The antiproliferative effects of benzoquinone/ascorbate combinations were significantly stronger than that of naphthoquinone/ascorbate combinations.”
Corrected as (new lines 165-167): “The antiproliferative effects of benzoquinone/ascorbate combinations were more pronounced than that of naphthoquinone/ascorbate combinations (Figure 2C-a,b,c versus Figure 2C-d,e,f).”
- Comment on line 185: “ThymoQ/A, DMBQ/A, MMBQ/A, CoQ0/A, K3/A, and BNQ manifested synergistic antiproliferative and cytotoxic effects. -> Cannot conclude without p-value calculation”.
Corrected as (new lines 187-193): “ThymoQ/A, DMBQ/A, MMBQ/A, CoQ0/A, K3/A, and BNQ manifested synergistic antiproliferative and cytotoxic effects (Figure 2D-b, c, d, e, h; Table S1 in the Supporting information). In this case, the synergism between quinone and ascorbate was well-expressed at concentrations between 1/100 and 5/500 μM/μM, but not at higher concentration (Figure 2D-l; Table S1). The effects of combinations of AtovaQ and prenylated quinones with ascorbate did not differ from that of the corresponding quinone administered alone (Figure 2D-f, g, j, k; Table 1S).”
- Comment on line 191: “The antiproliferative effects of CoQ1/A and K1/A were additive, whereas the effects of CoQ10/A and K2/A were rather antagonistic.”
Corrected as above (new lines 187-193).
- Comment on line 200: “AtovaQ/A and 200 BNQ/A demonstrated relatively weak but significant cytotoxicity to normal lymphocytes at a concentration of 20/2000 μM/μM (Figure 2F).”
Corrected as (new lines 201-203): “AtovaQ/A and BNQ/A demonstrated relatively weak but significant cytotoxicity to normal lymphocytes at a concentration of 20/2000 μM/μM (p<0.05) (Figure 2F).”
- Comment on lines 277-282: “The effects were synergistic, with synergism being more pronounced with K3/A than with CoQ0/A. AtovaQ/A significantly decreased the level of mitochondrial superoxide in Colon26 and MCF7 but its effect in U87MG cells was negligible (Figure 3H). No synergism was found between quinone and ascorbate.”
Corrected as (new lines 279-284): “CoQ0/A and K3/A were found to induce overproduction of mitochondrial superoxide in Colon26, MCF7, and U87MG cells (Figure 3H), while AtovaQ/A significantly decreased the level of mitochondrial superoxide in Colon26 and MCF7 but did not affect this parameter in U87MG cells. The effects of K3 and ascorbate administered alone were negligible, whereas CoQ0 alone significantly increased mitochondrial superoxide in all cancer cell lines.”
- Comment on line 496: “However, no synergism was observed between AtovaQ and ascorbate. This cannot be concluded without proper p-value calculation and labeling on the figure.”
We removed this sentence from the revised text. The effect of AtovaQ/A on cell proliferation was even antagonistic, compared to AtovaQ and ascorbate administered alone. Please, see Table S1 in the Supporting information.
- Reviewer's Comment: “Like in Figure 2, p-value calculation is completely missing in Figure 3.”
Answer: P-values were included in the charts of Figure 3A, B, C, D, H, I, and J. Please, see the revised Figure 3.
- Reviewer's Comment on line 236: “Effect was moderate. -> Define moderate. It seems like the effect is smaller than others, but still there. Again, difficult to tell without proper statistical calculations.”
Answer: Corrected. Please, see the new lines 244-245):
“CoQ1/A and BNQ/A also induced mitochondrial superoxide production, but the effect was less pronounced.”
- Reviewer's Comment on lines 253-258: “Specify which condition was used for calculation.”
“Very good correlations were established between the analysed parameters: cell proliferation/viability and mitochondrial superoxide (R=–0.8537; p<0.001) (Figure 3E); cell proliferation/viability and steady-state ATP (R=+0.8197; p<0.05) (Figure 3F); and mitochondrial superoxide and steady-state ATP (R=–0.6982; p<0.001) (Figure 3G).”
Answer: Corrected. Please, see the new lines 293-294:
“Correlation analysis between mitochondrial superoxide, ATP level, and cell proliferation/viability in Q/A-treated leukemic lymphocytes, presented in Figures 2D and 3A,C.”
Please, see the revised Legend of Figure 3.
- Reviewer's Comment on line 373: “…while K3/A caused only a growth arrest -> where did this conclusion came from? 4D? Also, p-value calculation needed.”
Answer: Corrected. Please, see the new lines 389-390:
“CoQ0/A decreased the size of the tumor, while K3/A caused only a growth arrest (p<0.05).” Please, see also revised Figure 4.
- Reviewer's Comment: “Indication of the number of technical/biological replicas in the figure legends is missing.”
Answer: Number of replicates and number of animals in each experimental group were included in the Figure legends. Please, see the revised Legends of Figures 2, 3 and 4.
Minor points
All minor points were also revised and corrected into the text.
They are mentioned in red in the revised version.

Round 2
Reviewer 1 Report
The authors consider all my suggestions contained in the first round of review.
The only issue that needs improvement is the quality of the all figures included in the manuscript - are illegible. I would like to suggest use a .TIFF format ect.
Author Response
Dear Reviewer,
High resolution figures (over 300 dpi) were submitted as JPEG-files to Mr. Ernest Goh (Assistant Editor) on April 19 via email.
Thank you for your constructive comments and positive decision about our manuscript!
With kind regards,
Rumiana Bakalova, PhD, DSci
(corresponding author)
Reviewer 2 Report
All of the reviewer's concerns were addressed.
Please correct:
Line 138: The aim of the present study was-> is
Author Response
Dear Reviewer,
Line 138 was corrected.
Thank you for your constructive comments and positive decision about our manuscript!
With best regards,
Rumiana Bakalova, PhD, DSci
(corresponding author)